# Investigation of bacterial community and histamine production in fresh mackerel at low temperature storage

Grace Margareta[1], Ayaka Nakamura[1], Kazumi Nimura[2], Motoyuki Yamazaki[2], Iori Oshima[2], Takashi Kuda[1], Hajime Takahashi[1]*

1 Department of Food Science and Technology, Tokyo University of Marine Science and Technology, Minato, Tokyo, Japan, 2 Shizuoka Prefectural Research Institute of Fishery and Ocean, Yaizu, Shizuoka, Japan

* hajime@kaiyodai.ac.jp

## Abstract

This study investigates microbial growth, community composition, and histamine production in freshly caught mackerel stored at 4°C and 10°C. Fish were obtained directly from fishermen and immediately placed in low-temperature storage to minimize external contamination. Microbial activity was assessed using total viable count, amplicon sequencing, and HPLC analysis. The total bacterial count exceeded 5 log CFU/mL after 5 days at 4°C and 2 days at 10°C, indicating the presence of histamine-producing bacteria (HPB), particularly under psychrophilic conditions. *Photobacterium* was identified as the dominant bacterial genus at both temperatures, suggesting its role as a key psychrophilic HPB. Histamine levels increased progressively, reaching 5.11±2.43 ppm at 4°C and 11.18±4.67 ppm at 10°C, while histidine content declined. Despite cold storage and strict contamination control, naturally occurring bacteria in fish continued to grow and produce histamine. These findings highlight the importance of temperature control in preventing histamine accumulation and provide new insights into microbial dynamics and food safety risks in fresh mackerel.

## 1. Introduction

The fisheries industry plays an important role in food consumption, especially in Japan. However, fish might be contaminated and lead to food-borne illness such as scombroid poisoning caused by high levels of histamine [1]. According to data obtained until 2020 from the Rapid Alert System for Food and Feed, countries such as Italy, France, Germany, Spain, and Belgium have reported the presence of histamine in fish and fishery products. This might be due to the consumption of dark muscle fish containing free L-histidine, which can be enzymatically converted to histamine [2]. Moreover, according to the European Union, there has been an upward trend in

**Data availability statement:** All relevant data are within the manuscript.

**Funding:** The author(s) received no specific funding for this work.

**Competing interests:** The authors have declared that no competing interests exist.

histamine poisoning outbreaks several outbreaks, with a total of up to 488 human cases and 115 hospitalizations in 2018 [3].

In most cases, mesophilic bacteria are the main causative agents of Histamine Food Poisoning (HFP) [4]. However, the possibility of histamine production by psychrophilic histamine producing bacteria (HPB) should be explored. Refrigerated storage is the most common technique to prevent the proliferation of microorganisms associated with histamine formation. Yet, temperature abuse is unlikely to be the only factor responsible for promoting HFP. To date, scombroid poisoning outbreaks have been associated with psychrophilic HPB such as *Photobacterium phosphoreum* and *Morganella psychrotolerans*, which are known to produce harmful concentrations of histamine in chilled fish [5,6,7]. The growth and production of histamine by psychrophilic HPB are unpredictable, as both timing and temperature controls could be insufficient to control these bacteria and the formation of histamine (>500 ppm) at low-temperature storage indicates a pathogenic role.

Fish catch goes through many processes from harvest to consumption. There is a possibility of poor handling, delayed evisceration, and temperature fluctuations leading to increased microbial contamination, spoilage, and reduced fish quality at each stage [8,9]. In this study, mackerel was selected as a model food to investigate the behavior of initially attached HPB. The fish samples were purchased from the fisherman and directly stored, reflecting the authentic representation of early stages and actual microbial presence which offers insights into the natural ecosystem and its impact on bacterial attachment and histamine formation, providing a deeper understanding of microbial dynamics in fresh fish.

This study aimed to analyze bacterial counts, community composition, and histamine production by naturally occurring bacteria in fresh mackerel stored at low temperatures. Unlike previous studies focusing on market fish, this is the first study to use fresh mackerel directly harvested by fishermen, providing data with minimal external contamination in order to provide a better insight into the intrinsic bacterial dynamics of fresh fish during cold storage. In addition, the research investigates whether psychrophilic bacteria, known for histamine production at low temperatures, are predominant at the end of the storage period by specifically using cold-chain scenarios temperature.

## 2. Materials and methods

### 2.1 Sample preparation

Fresh mackerel were obtained from local fishermen (boat) in Shizuoka prefecture (Japan) and transported directly to the Food Microbiology Laboratory (Tokyo University of Marine Science and Technology, Japan) on the same day. Six whole mackerel were used and placed in sterile iced polystyrene boxes (0ºC), coated with newspaper as insulator, and each fish was covered with transparent plastic. Immediately upon arrival at the laboratory, the fish were aseptically divided into two storage temperature groups: three fish were stored at 4°C and three fish at 10°C. Fish samples were stored in temperature-controlled incubators set at 4ºC and 10ºC. The mackerel was

observed on days 0, 3, and 5 for the 4ºC storage, and on days 0, 1, 2, and 3 for the 10ºC storage. The 4°C condition was selected to represent typical domestic refrigeration temperatures, while 10°C was chosen based on regulatory guidelines for raw fish storage temperatures in Japan [10]. The experiment was repeated four independent times to ensure reproducibility.

## 2.2 Bacterial growth

The total viable bacteria (TVC) method was used to analyze total bacteria growth with the range of 25–250 colonies on a plate [11]. The mackerel was cut into five pieces, each measuring 5 × 5 cm, collected from different regions of the fish body, with a total weight of approximately 25 g. All pieces were homogenized in 225 mL of L-histidine broth, which contained 10 g of Bacto Peptone (Gibco, USA), 3 g of Bacto Yeast Extract (Gibco, USA), 5 g of D-glucose (Kokusan Chemical Co., Ltd., Japan), 5 g of L-histidine (Fujifilm, Japan), 500 mL of Artificial Sea Water (ASW), and 500 mL of distilled water. The ASW was prepared using 17.55 g of NaCl, 0.75 g of KCl, 0.285 g of $Na_2SO_4$, 5.1 g of $MgCl_2 \cdot 6H_2O$, and 0.192 g of $CaCl_2 \cdot 2H_2O$ (all from Kokusan Chemical Co., Ltd., Japan), dissolved in 1000 mL of distilled water. Subsequent serial dilutions were performed using saline solution composed of 8.5 g of NaCl dissolved in 1000 mL of distilled water. We continued with predetermined dilution in saline solution , growth in Tryptone Soy Agar (TSA) with 50% ASW (TSA powder (Nippon Becton Dickinson, Japan) 40 g, 500 mL ASW and distilled water 500 mL per 1000 mL), and incubated at 15ºC for 48–72 h and at 30ºC for 24–48 h. Incubation at 15°C was intended to monitor the growth of psychrophilic bacteria, whereas incubation at 30°C was used to support the growth of mesophilic bacteria.

The most probable number (MPN) method was used with L-histidine broth medium. Each test tube from the MPN assay was subsequently analyzed for histamine production using paper chromatography, in order to confirm whether the bacterial colonies present were capable of histamine synthesis [7]. A 0.5 µL aliquot from each sample was spotted onto Whatman No. 1 filter paper (diameter = 18 cm) and dried using a hairdryer. Chromatographic separation was performed using a solvent system consisting of butanol and ammonium hydroxide solution in a 1:1 ratio. After development, the paper was dried and sequentially sprayed with two reagents: Reagent A (0.5% sulfanilic acid in 0.1% HCl mixed with 1% $NaNO_2$ in a 1:1 ratio) and Reagent B (saturated $Na_2CO_3$ solution). These reagents react with amine functional groups to yield a visible colored spot indicative of histamine presence. The samples were incubated under the same conditions as mentioned earlier. The bacterial growth observed on TSA with ASW represents the naturally occurring microbiota of mackerel, whereas the bacteria detected using the MPN method were classified as suspected HPB.

## 2.3 High-throughput sequencing

For bacterial community analysis, 1 mL portion from 225mL histidine broth and 25g fish in the sample bag was collected on the last day of storage. The homogenized samples were centrifuged at 8,000 × g for 10 minutes to separate the supernatant, from which genomic DNA was subsequently extracted using the NucleoSpin® Tissue Extraction Kit (Takara Bio, Japan), following the manufacturer's protocol. The samples were then subjected to amplicon sequencing targeting the V3–V4 region of the 16S rRNA gene using the primer set as shown in the Table 1. Sequencing was performed using the Illumina MiSeq Reagent Kit v3 with 2 × 300 bp paired-end reads at Bioengineering Lab. Co., Ltd. (Kanagawa, Japan). Negative controls (extraction blanks) were included to monitor contamination. The raw sequences were processed and the alpha diversity indices were estimated and analyzed using the QIIME 2.0 (DADA2 model).

## 2.4 Determination of histamine production

Briefly, 10 g of mackerel was blended in 90 mL of sterile distilled water and homogenized. Subsequently, 1 mL of the homogenate was mixed and taken for preliminary histamine detection using a Check color Histamine kit (Kikkoman Corp., Chiba, Japan), following the manufacturer's instructions with volume modification.

**Table 1. Primer set for V3–V4 region of the 16S rRNA gene.**

| Primer name | Sequence (5'→3') |
| --- | --- |
| 1st 341f_mix* | ACACTCTTTCCCTACACGACGCTCTTCCGATCT-NNNNN-CCTACGGGNGGCWGCAG |
| 1st 805r_mix* | GTGACTGGAGTTCAGACGTGTGCTCTTCCGATCT-NNNNN-GACTACHVGGGTATCTAATCC |
| 2nd 341f | AATGATACGGCGACCACCGAGATCTACAC-Index2-ACACTCTTTCCCTACACGACGC |
| 2nd 805r | CAAGCAGAAGACGGCATACGAGAT-Index1-GTGACTGGAGTTCAGACGTGTG |

*To improve the sequence analysis quality, mixed primers with random sequences of 0–5 bases in length are used.

Histamine and histidine were determined using the High-Performance Liquid Chromatography (HPLC) method according to the procedures of [12]. For the HPLC sample preparation, 1 mL of the homogenized sample was boiled for 15 minutes to denature proteins, followed by centrifugation at 8,000 × g for 10 minutes. The resulting supernatant was filtered through a 0.45 µm membrane filter. The HPLC system (Shimadzu Corporation, Japan) consist of an auto-sampler (SIL-20 AC HT), pump (LC-20AD), degasser (DGU-20A SR), column oven (CTO-20A), and UV–Vis detector (SPD-20AV), with chromatographic data analyzed using LabSolutions software (version 5.92). Separation was performed on an ODS-II column (STR ODS-II, 250 × 4.6 mm, 5 µm particle size; Shinwa Chemical Industries Ltd., Japan) maintained at 40°C.

Gradient elution was performed with 5% ethanol in a 150 mM sodium acetate buffer at pH 6.0 (sol. A) and with 60% acetonitrile in water (sol. B). The linear-gradient program for the separation procedure was from 0% of sol. B to 45% of sol. B in 30 min at a flow rate of 1.0 ml/min. The wavelength of the variable-wavelength detector was set at 420 nm. Calibration curves were constructed using histamine and histidine standards at concentrations ranging from 0.1 to 50 mg/L, with $R^2$ values exceeding 0.995. Quantification was based on peak area comparison with external standards. All HPLC analysis was conducted by the Shizuoka Prefectural Research Institute of Fisheries and Ocean, Japan.

## 2.5 Statistical analysis

All measurements were conducted in quadruplicate and the data are expressed as the mean ± standard error (SE). The data was analyzed and evaluate with One-Way ANOVA (Single Factor) and correlation analysis to assess the relationship between bacterial genera and histamine levels, conducted using Microsoft Excel Office 16. Differences were considered significant at the $p < 0.05$.

## 3. Results and discussion

### 3.1 The change in bacterial count

The bacterial count changes are shown in Fig 1a,1b. In Japan, the Ministry of Health, Labor and Welfare sets acceptable TVC limits depending on the fish products, with a maximum value for frozen fish for raw consumption at 5 log CFU/g [13]. At 4°C storage, bacterial counts reached over 5 log CFU/g after 5 days, with 5.84 ± 0.27 log CFU/g for psychrophilic bacteria and 5.76 ± 0.26 log CFU/g for mesophilic bacteria. At 10°C storage, bacterial counts reached more than 5 log CFU/g within 2 days incubation time, with the final number of psychrophilic bacteria at 6.28 ± 0.15 log CFU/g and mesophilic bacteria at 6.33 ± 0.17 CFU/g. However, in this study, there was no significant difference between the psychrophilic and mesophilic bacteria growth in each temperature treatments ($p > 0.05$). The faster bacterial growth at 10°C storage compared to 4°C suggests that temperature significantly influences the growth rate of bacteria ($p < 0.05$). Based on their optimal growth temperature, psychrophilic bacteria are adaptable to low temperatures

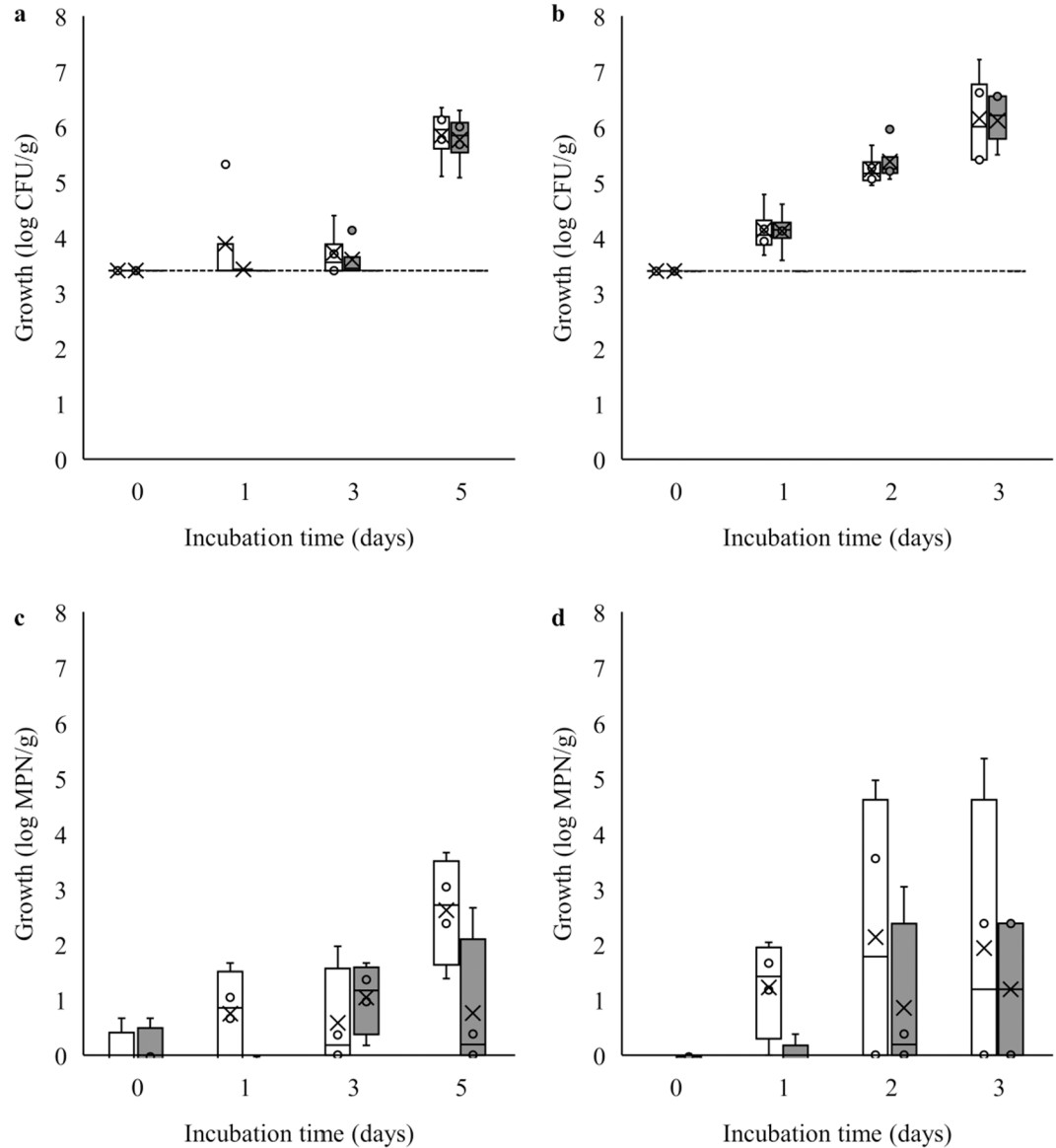

**Fig 1. Total viable count and most probable number of suspected HPB in Mackerel at low temperature storage.** Total viable count of bacteria in Mackerel during (a) 4°C and (b) 10°C storage and most probable number of suspected HPB in Mackerel during (c) 4°C and (d) 10°C storage. Bacterial counts were evaluated in two conditions: 15°C as psychrophilic conditions (white color) and 30°C as mesophilic conditions (gray color). The reliability detection limit set in this study for TVC (a,b) was 3.4 log CFU/ g, whereas for MPN (c,d) was 0 MPN/ 10 g.

below 20°C, while mesophilic bacteria thrive between 20–45°C. However, it is possible that mesophilic bacteria can adapt to lower temperatures [14]. Storage at 10°C is closer to the optimal growth temperature for bacteria, allowing rapid growth and providing suitable conditions for their metabolic activity. A study on mackerel collected from the fishing harbor in Southern Taiwan showed a similar trend, where the aerobic plate count reached up to 7.1 log CFU/g after 5 days stored at 4°C and approximately 9 log CFU/g when stored at 15°C for 3 days [15] higher than those reported in this study. This indicates potential bacterial contamination resulting in higher bacterial counts compared to the fresh samples used in this study.

The suspected HPB growth results using the MPN method are shown in Fig 1c and 1d with a lower limit of 0 log MPN/g. HPB was detected in all samples; yet the numbers were lower compared to the total bacterial count. For suspected psychrophilic HPB, the final sample bacterial count stored at 10°C was higher than that stored at 4°C, with approximately 4.06 log MPN/g after 3 days and 2.17 log MPN/g after 5 days, respectively. In contrast, for suspected mesophilic HPB, the number of bacteria observed in the tube was less, with a final sample count of approximately 1.36 log MPN/g for 4°C and 1.38 log MPN/ g for 10°C. At 10°C, a reduction in mesophilic HPB was also observed at the end of storage.

From the total bacteria detected, several of the bacterial communities comprised HPB. Interestingly, in this study, the MPN showed that psychrophilic HPB were dominant, which were able to grow, survive, and produce histamine during the cold storage period. The differences in growth rates between psychrophilic and mesophilic bacteria can be attributed to their respective temperature preferences. This is consistent with the findings of a study on frozen swordfish [7], where HPB was observed to grow with a final count equal to or lower than the total aerobic bacteria, also showed a tendency to grow for >30 hours at 4, 10, and 15°C for the psychrophilic HPB.

### 3.2 Composition of bacterial community in the last storage

Table 2 shows the diversity analysis of the two low temperature treatments. At 10°C, a higher number of Amplicon Sequence Variant (ASV) were observed compared to those at 4°C. Statistical analysis indicated a significant difference ($p<0.05$) between the two temperature treatments, indicating a greater diversity of microbes in the samples at 10°C and a richer and more complex microbial community structure. Storage at 10°C falls within the optimal growth temperature range for various bacterial species, providing favorable conditions for their proliferation and colonization, resulting in increased bacterial richness. This discrepancy implies that the 10°C environment is more conducive to bacterial growth and colonization, fostering a greater diversity of bacterial species. Some species may thrive more effectively at 10°C, while others may prefer the 4°C environment, leading to differences in species composition between the two temperatures. At lower temperatures, organisms face limitations in nutrient supply, hampering their growth rates due to reduced substrate affinity. Sub-optimal temperatures render microorganisms unable to effectively absorb nutrients from their surroundings, exacerbating near-starvation conditions [16]. Simpson and Shannon indices were used to assess bacterial diversity and evenness. As for Simpson and Shannon diversity index, the 10°C treatment showed slightly lower and higher values, respectively. A higher Simpson index indicates lower diversity due to dominance by a few species, while a higher Shannon index reflects greater diversity and a more even distribution. However, statistically, no significant differences ($p>0.05$) were observed in either the Simpson or Shannon values between the two treatments, suggesting that the both low temperature treatments did not affect the diversity of the samples.

The genus abundance presents on the samples on the last day of storage is shown in Fig 2. Bacterial species comprising less than 1% are not included in this graph. Across four repeated experiments in both temperature treatments, the most abundant genus was *Photobacterium*. *Photobacterium* genera are naturally seawater inhabitants and are widely documented to be capable of producing histidine decarboxylase, making them the main bacterial genus involved in histamine production in psychrophilic environments. Some psychotropic *Photobacterium* species isolated from fish produce

**Table 2. Diversity indices in Mackerel at the end day of low temperature storage.**

| Treatment | 4°C | 10°C |
|---|---|---|
| 16S rDNA (V3/V4) amplicon sequence analysis | | |
| Number of ASV | 18.75±5.1[a] | 46.25±3.22[b] |
| Shannon index (*H'*) | 1.97±0.13[a] | 2.24±0.32[a] |
| Simpson diversity (*D*) | 0.80±0.04[a] | 0.79±0.06[a] |

Mean values with different letters within a row are significantly different ($p<0.05$).

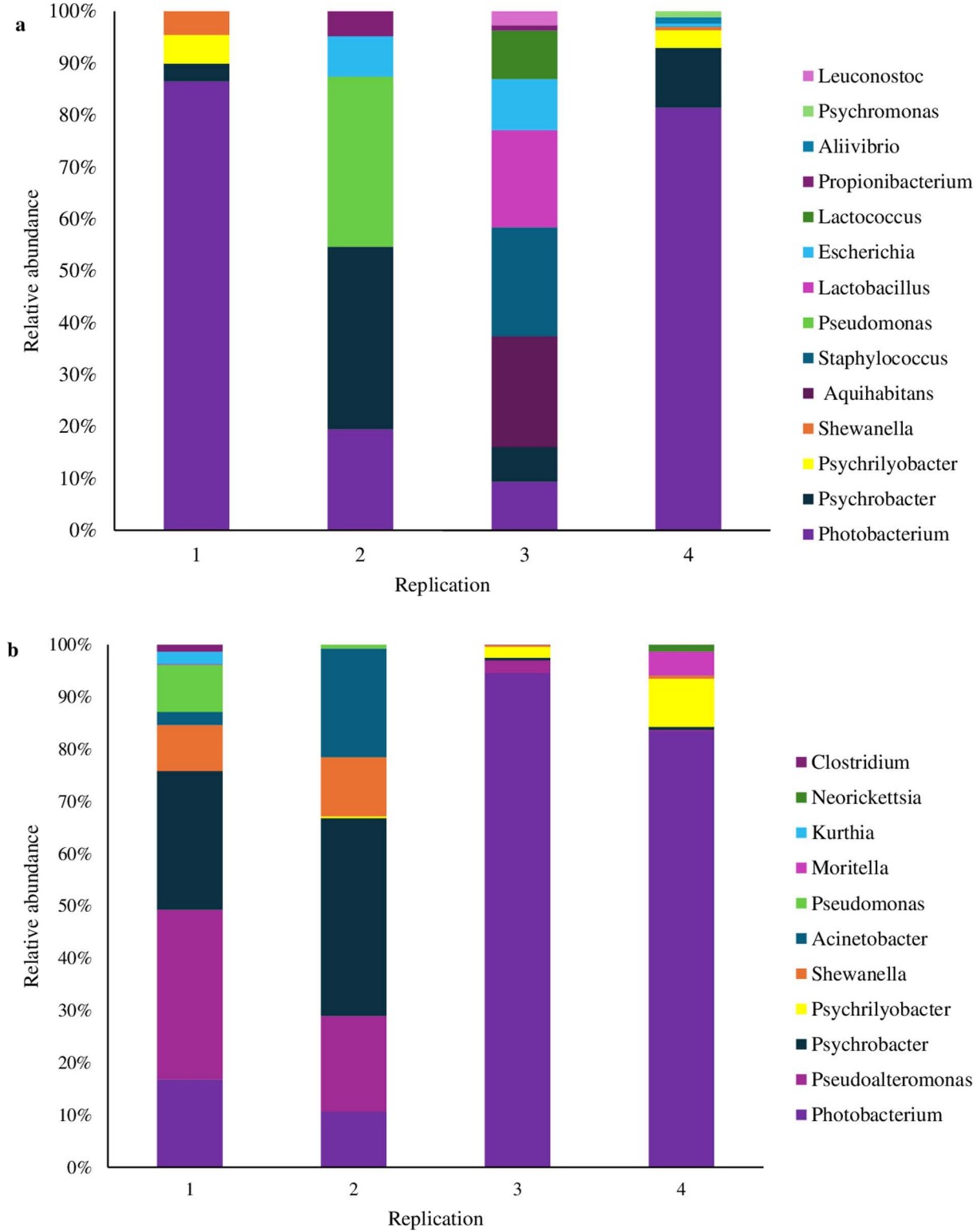

**Fig 2. Relative abundance of bacteria community in genus level.** Relative abundances of dominant genera assigned to V3/V4 region of 16S rRNA sequences detected in mackerel which was stored at (a) 4°C and (b) 10°C for 5 and 3 days, respectively. Scale in y axis reflects the normalized abundance percentages (%).

toxic concentrations of histamine [17], and certain *Photobacterium* strains are associated with Scombroid food poisoning [6]. A study [18] detected several *Photobacterium* species that significantly produce histamine (>200 ppm) such as *P. aquimaris, P. kishitanii, P. damselae,* and *P. phosphoreum.* The *hdc* gene was detected in all of them except *P. phosphoreum.*

### 3.3 Histamine production during cold storage

The histamine content of samples from HPLC analysis is described in Table 3. Histamine levels in mackerel showed an increase throughout the cold storage period, both at 4°C and 10°C. Between the two temperature treatments, mackerel stored at 10°C showed a higher and more accelerated increase in histamine production. The final histamine content reached 5.11±4.86 ppm within 5 days at 4°C storage and 11.18±9.34 within 3 days at 10°C storage. These results show similar trends to a study [19], where histamine levels in Pacific Mackerel at 15°C storage increased to 14 mg/100 g (approximately 140 ppm) in only two days, while at 4°C, histamine was not produced or detected <1 mg/100 g until the sixth day, after which it began to increase and accumulated to 57.4 mg/100 g (approximately 574 ppm) by the fourteenth day. The histamine content in mackerel ranges from 2.7 to 245.8 mg/kg (equivalent to ppm) [20], indicating that the results of this study fall within that range.

In addition to histamine content, the histidine content in fresh mackerel during storage was also measured, as shown in Table 3. Interestingly, in this study, the measured histamine and histidine values showed opposite trends: the longer the storage time, the more the amount of histamine accumulated, while the detection of histidine decreased. Histamine formation by bacteria is influenced by the presence of free histidine, which serves as a substrate for histidine decarboxylase. Bacterial autolysis or protein digestion can lead to faster histidine release from tissue proteins, thereby increasing histamine formation [21]. In this study, the HPLC results indicated that free histidine in mackerel was produced by bacterial enzymes and converted into histamine. Similar observations of histidine and histamine content in mackerel were reported by [22], which stated that histidine content decreased during storage, with some of it catalyzed into histamine.

The recommended minimal histidine level for histidine decarboxylase function ranges from 100 to 200 mg/100 g [23]. Factors such as dietary intake, weather conditions, sex, and age stages of fish influence the free histidine levels in fish tissues. Based on recommendations, a histamine level of ≥ 50 ppm in fish tissue is considered indicative of spoilage, and if it is ≥ 500 ppm, it into the range that poses a threat to human health [24,25]. Although the histamine levels detected in this study are far below the histamine regulatory standards, some individuals with low tolerance to histamine can still experience symptoms of toxicity. Lower levels may also cause symptoms in individuals with reduced histaminase activity, such as monoamine oxidase and diamine oxidase [26]. Certain medical conditions, including the use of anti-tuberculosis drugs, are known to inhibit enzymes that can cause histamine toxicity. A report from October 2, 2003, where eight patients experienced symptoms of histamine intoxication such as flushing, headache, hives, palpitations, wheezing, and shortness of breath that occurred within 20 min to 2 h after consuming powdered saury paste and omelet with detectable histamine concentrations reaching 32 mg/100 grams [27].

In this study, the production of histamine in fresh mackerel was affected by psychrophilic histamine-producing bacteria. Correlation analysis was performed to evaluate the relationship between microbial diversity and histamine levels at the final storage

**Table 3. Histamine evaluation of Mackerel during low temperature storage by HPLC.**

| Temp (°C) | Time (days) | Histidine (ppm) | Histamine (ppm) |
|---|---|---|---|
| 4 | 0 | 586.13±47.62 | 0.24±0.04 |
| | 1 | 557.92±108.74 | 0.30±0.13 |
| | 3 | 487.52±103.31 | 0.42±0.13 |
| | 5 | 476.68±84.54 | 5.11±2.43 |
| 10 | 0 | 586.13±47.62 | 0.24±0.04 |
| | 1 | 515.21±108.44 | 1.64±0.12 |
| | 2 | 421.10±37.85 | 5.11±2.39 |
| | 3 | 405.32±47.79 | 11.18±4.67 |

time point and showed that *Photobacterium* as the most abundance genus of both temperature treatments showed a positive correlation with accumulated histamine concentration (S1 Fig). Previous studies have shown variability in histamine production capacity among different *Photobacterium* strains, including differences in gene expression and metabolic activity under cold storage [18,7]. The naturally occurring bacteria grew more rapidly when stored at 10°C, reaching more than 5 log CFU/g after only 2 days of storage due to the favorable conditions at 10°C compared to the 4°C storage. Storage at both temperatures also showed a high presence of psychrophilic bacteria due to the cold temperature range used. Bacterial community analysis showed that the dominant natural bacterial genera at the end of storage were *Photobacterium*. Histamine was detected in mackerel during storage and increased over time, reaching 5.11±2.43 ppm and 11.18±4.67, at 4°C and 10°C storage, respectively. Concurrently, the histidine content decreased, indicating that free histidine in fish was converted into histamine.

At both temperatures, there was an increase in the total bacterial count, leading to histamine accumulation in the mackerel. This may be due to the growth of suspected HPB, identified as *Photobacterium* species, which can convert histidine in fish into histamine. This finding indicates that storage at 10°C promotes more rapid bacterial growth compared to 4°C, which is associated with a higher accumulation of histamine, further supporting the correlation between histamine production and microbial activity. This study provides novel insights into the natural bacterial dynamics and histamine production in freshly harvested mackerel stored at low temperatures, emphasizing the critical role of strict temperature control in mitigating histamine accumulation and enhancing the microbial safety of seafood.

## Supporting information

**S1 Fig. Correlation analysis between histamine level and microbial flora at genus level.** Correlation matrix of 18 significantly abundant genera and histamine level. The matrix displays positive and negative correlation between genera and histamine accumulation, represented by red and blue colors, respectively. Correlation values are normalized between −1 and +1, as indicated by the color scale bar below.
(PDF)

## Acknowledgments

We would like to thank *Editage* (www.editage.com) for English language editing.

## Author contributions

**Conceptualization:** Kazumi Nimura, Hajime Takahashi.

**Data curation:** Grace Margareta.

**Investigation:** Grace Margareta, Ayaka Nakamura, Kazumi Nimura, Hajime Takahashi.

**Methodology:** Grace Margareta, Ayaka Nakamura, Motoyuki Yamazaki, Iori Oshima, Hajime Takahashi.

**Resources:** Kazumi Nimura, Motoyuki Yamazaki, Iori Oshima, Takashi Kuda, Hajime Takahashi.

**Supervision:** Hajime Takahashi.

**Writing – original draft:** Grace Margareta.

**Writing – review & editing:** Hajime Takahashi.

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
