## [Decision Letter · Decision Letter 0]

13 May 2025

Dear Dr. Takahashi,

Thank you for submitting your manuscript to PLOS ONE. After careful consideration, we feel that it has merit but does not fully meet PLOS ONE’s publication criteria as it currently stands. Therefore, we invite you to submit a revised version of the manuscript that addresses the points raised

We look forward to receiving your revised manuscript.

Kind regards,

Satheesh Sathianeson, Ph.D

Academic Editor

PLOS ONE

Journal Requirements:

Reviewers' comments:

Reviewer's Responses to Questions

**Comments to the Author**

1. Is the manuscript technically sound, and do the data support the conclusions?

Reviewer #1: Yes

Reviewer #2: Partly

2. Has the statistical analysis been performed appropriately and rigorously?

Reviewer #1: Yes

Reviewer #2: No

3. Have the authors made all data underlying the findings in their manuscript fully available?

Reviewer #1: Yes

Reviewer #2: No

4. Is the manuscript presented in an intelligible fashion and written in standard English?

Reviewer #1: Yes

Reviewer #2: No

Reviewer #1: The manuscript presents the results of a study aimed understanding the insights of histamine formation in fish. Results quite interesting and may contribute to related literature. However, the paper needs some improvements listed below.

1. Please cite a reference for the total viable method.

2. Line 81: Sampling procedure is not clear. Were five pieces (totally 25 g) cut and homogenized in 225 ml culture medium? Please add and explanation.

3. Line 81-87: The composition of L-histidine broth is hardly understandable. Please clear. You may give the composition of artificial sea water in a separate sentence.

4. Lines 178-181: What are the means of Shannon and Simpson diversity indexes? Do temperature affect the diversity? Please discuss.

5. Figure 2: There are significant differences between the replications in respect to microbial diversity. How could you explain this situation? Are these differences significant or expected? Please interpret these results in detail.

6. The conclusion parts needs to be improved. The last sentence (lines 250-251) has no contribution, since it has already been well known that the histamine production is related to microbial activity. Please mention that how these result can contribute to the safety of fish in respect to histamine.

7. The reference numbered 12 is too old. Please cite a recent reference.

Reviewer #2: General Comments

Line 70: The manuscript does not provide the number of mackerel samples used in the study and does not specify how they were distributed among storage temperature and time groups. Information on sample size and replication is critical for assessing experimental design, statistical validity, and reproducibility.

Line 79: The authors state that total viable counts were performed with TSA agar. However, it is not clear from the text whether mesophilic, psychrophilic or histamine producing bacteria (HPB) were counted. It is stated that “HPB colonies were observed” but this is not supported by a specific identification method.

The incubation temperatures used are complicated and unclear: Incubation temperatures are given on TSA agar: 4°C, 20°C, 30°C. However, which group of bacteria do they target? For example: 4°C → psychrophilic 20–30°C → mesophilic

This relationship is not explained and it is not stated why these temperatures were chosen.

No selective media for histamine-producing bacteria (HPB): If HPB identification is to be made, selective media such as Niven’s agar, decarboxylase media would be required.

HPB cannot be identified with TSA alone, this is scientifically incorrect.

Line 95-120, for all material and methods: PCR/amplicon sequencing part: "Amplicon sequencing" was performed, but which gene region was targeted (16S V3-V4 is understood from the table, but it is not clear in the text), which primer set, which control samples were used, where is the raw data? The reader cannot clearly understand which bacteria were defined as HPB and on what basis. The methodology used for counting psychrophilic and mesophilic bacteria is not clear. TSA + ASW environment and temperatures are given, but which bacterial groups were classified as psychrophilic/mesophilic and on what basis is not explained.

HPLC method: Reference is provided, but separation conditions, detector wavelength, calibration curves, sample preparation steps are quite superficial.

Aim: The novelty aspect is not emphasized enough: How is this study different from previous studies on histamine production? For example, was a new HPB strain identified at the metagenomic level?

Statistical analysis: The statistical analysis is limited to ANOVA. Correlation analyses between microbial diversity and histamine levels could have been performed. Moreover, the type of variance analysis applied is not specified in the manuscript.

Figures and tables: The figures and tables are overly simplistic and not reader-friendly. For instance, Figure 2 presents only genus-level relative abundances; there is no specific analysis at the species level or of HPB carrying the hdc gene.

The manuscript should be thoroughly revised and rewritten by incorporating the above suggestions to present a clear, methodologically, and scientifically valuable study. Without these substantial corrections, the manuscript remains too ambiguous, inconsistent, and incomplete to be considered for publication in its current form.

**Do you want your identity to be public for this peer review?** For information about this choice, including consent withdrawal, please see our Privacy Policy

Reviewer #1: No

Reviewer #2: No

---

## [Author Response · Author response to Decision Letter 1]

27 Jun 2025

General Comments

Line 70: The manuscript does not provide the number of mackerel samples used in the study and does not specify how they were distributed among storage temperature and time groups. Information on sample size and replication is critical for assessing experimental design, statistical validity, and reproducibility.

Thank you for your feedback, we have added more detailed information regarding sample preparation on Line 76-85.

Line 79: The authors state that total viable counts were performed with TSA agar. However, it is not clear from the text whether mesophilic, psychrophilic or histamine producing bacteria (HPB) were counted. It is stated that “HPB colonies were observed” but this is not supported by a specific identification method.

The incubation temperatures used are complicated and unclear: Incubation temperatures are given on TSA agar: 4°C, 20°C, 30°C. However, which group of bacteria do they target? For example: 4°C → psychrophilic 20–30°C → mesophilic

This relationship is not explained and it is not stated why these temperatures were chosen.

No selective media for histamine-producing bacteria (HPB): If HPB identification is to be made, selective media such as Niven’s agar, decarboxylase media would be required.

HPB cannot be identified with TSA alone, this is scientifically incorrect.

Thank you for the comments! This study employed four different temperature conditions: two for cold storage of the fish (4°C and 10°C) and two for incubation of bacterial cultures (15°C and 30°C). We add the cold treatments explanation in Line 82-85 and the bacteria incubation explanation in Line 98-100. In this study, cold storage temperatures were applied to evaluate the quality of mackerel during storage in terms of bacterial counts, bacterial diversity, and histamine accumulation. Meanwhile, the incubation temperatures were used to compare the survival and growth of psychrophilic and mesophilic bacteria that persisted throughout the storage period.

Selective media were not used for bacterial enumeration in this study, as the primary objective was not to specifically isolate histamine-producing bacteria, but rather to assess whether such bacteria are dominant among the natural bacterial population present on fresh mackerel. It was clearly established that the bacteria cultured on Tryptic Soy Agar (TSA) represent the naturally occurring microbiota of mackerel, not exclusively histamine-producing strains. In contrast, for the Most Probable Number (MPN) method, L-histidine was used as the primary substrate for histamine production. Bacterial isolates that successfully grew in this medium and subsequently tested positive for histamine production using paper chromatography were classified as suspected histamine-producing bacteria. This explanation added in Line 101-113.

Line 95-120, for all material and methods: PCR/amplicon sequencing part: "Amplicon sequencing" was performed, but which gene region was targeted (16S V3-V4 is understood from the table, but it is not clear in the text), which primer set, which control samples were used, where is the raw data?

Thank you for the detailed questions. In this study, the microbial community analyses using 16S rRNA sequencing with the targeted gene region V3–V4 region with binding sequence 410 bp and lead length of 280 bp. The primers for amplifying using a two-step tailed PCR are 341f and 805r with the detail in Table 1. As for the control samples, Negative control is prepared using nuclease-free water as a template. The revised method version was added in the Line 116-126. Raw data will be made available upon request.

The reader cannot clearly understand which bacteria were defined as HPB and on what basis. The methodology used for counting psychrophilic and mesophilic bacteria is not clear. TSA + ASW environment and temperatures are given, but which bacterial groups were classified as psychrophilic/mesophilic and on what basis is not explained.

Thank you for your concern regarding the classification basis of HPB! However, in this section, the specific identification of HPB was not conducted. Overall, the primary objective of this study was not to guide readers toward the precise identification of HPB, but rather to describe the naturally occurring bacterial community present in fresh mackerel during cold storage. Indeed, the results of this study showed that the dominant genus was Photobacterium, which has been previously identified in the literature as a psychrophilic bacterium responsible for histamine production. Therefore, based on both prior research and the findings of this study, Photobacterium is considered the primary suspected psychrophilic HPB.The methods for enumerating psychrophilic and mesophilic bacteria, as well as the use of TSA medium supplemented with ASW, have been explained and added in the methodological section above (Line 98-100).

HPLC method: Reference is provided, but separation conditions, detector wavelength, calibration curves, sample preparation steps are quite superficial.

Thank you for your toughful comments! We have provided more technical information regarding the HPLC method on Line 134-149.

Aim: The novelty aspect is not emphasized enough: How is this study different from previous studies on histamine production? For example, was a new HPB strain identified at the metagenomic level?

Thank you for your valuable comment. We appreciate your interest in clarifying the novelty of our study. While the identification of a new HPB strain at the metagenomic level was not the primary focus of this work, our study offers several novel contributions:

• Minimization of external contamination

Unlike many previous studies that analyzed fish samples from retail or distribution points, our study used freshly caught mackerel, allowing us to investigate the naturally occurring bacterial community under controlled conditions with minimal external contamination. This approach provides clearer insights into the intrinsic bacterial dynamics of fresh fish during storage.

• Temperature-specific focus on psychrophilic HPB

We specifically explored bacterial behavior at 4°C and 10°C to reflect realistic cold-chain scenarios and to evaluate the role of psychrophilic bacteria in histamine accumulation. This temperature-focused analysis is less emphasized in prior literature, which typically assesses mesophilic HPB under broader or higher temperature ranges.

We have stated the novelty point in the cover letter and the manuscript to better emphasize these aspects of novelty and distinction from previous studies in Line 67-71.

Statistical analysis: The statistical analysis is limited to ANOVA. Correlation analyses between microbial diversity and histamine levels could have been performed.

Thank you for your valuable input. We have included a correlation analysis between bacteria number and histamine levels in Line 272-278 with a supplementary figure (S1_fig).

Moreover, the type of variance analysis applied is not specified in the manuscript.

We have now clarified in the revised manuscript that One-Way ANOVA (Single Factor) was used to analyze the differences in alpha diversity metrics (Shannon index, Simpson index, number of ASVs, and total read counts) between the 10°C and 4°C storage conditions in Line 153-155.

Figures and tables: The figures and tables are overly simplistic and not reader-friendly. For instance, Figure 2 presents only genus-level relative abundances; there is no specific analysis at the species level or of HPB carrying the hdc gene.

Thank you for your input, the reason we did not include species-level classification in this study is because amplicons sequencing targets fragments less than or equal to 400 bp of the 16S rRNA gene (V3-V4 region), which does not provide sufficient resolution for accurate and reliable species-level identification. For reference, previous studies have shown that 16S rRNA sequencing with short reads can lead to misclassification at the species level due to high sequence similarity among closely related taxa (Johnson et al., 2019; Janda & Abbott, 2007). Thus, we limited our analysis to the genus level to ensure greater confidence and accuracy in the interpretation of microbial community composition.

The manuscript should be thoroughly revised and rewritten by incorporating the above suggestions to present a clear, methodologically, and scientifically valuable study. Without these substantial corrections, the manuscript remains too ambiguous, inconsistent, and incomplete to be considered for publication in its current form.

Thank you once again for your time and professional review work on our manuscript. If there are any other modifications we could make, we would like very much to modify them and we really appreciate your help. We look forward to your feedback on the revised version.

References:

Janda, J. M., & Abbott, S. L. (2007). 16S rRNA gene sequencing for bacterial identification in the diagnostic laboratory: Pluses, perils, and pitfalls. Journal of Clinical Microbiology, 45(9), 2761–2764. https://doi.org/10.1128/JCM.01228-07

Johnson, J. S., et al. (2019). Evaluation of 16S rRNA gene sequencing for species and strain-level microbiome analysis. Nature Communications, 10, 5029. https://doi.org/10.1038/s41467-019-13036-1

---

## [Decision Letter · Decision Letter 1]

29 Jul 2025

Dear Dr. Takahashi,

Thank you for submitting your manuscript to PLOS ONE. After careful consideration, we feel that it has merit but does not fully meet PLOS ONE’s publication criteria as it currently stands. Therefore, we invite you to submit a revised version of the manuscript that addresses the points raised during the review process.

We look forward to receiving your revised manuscript.

Kind regards,

Satheesh Sathianeson, Ph.D

Academic Editor

PLOS ONE

Journal Requirements:

Additional Editor Comments:

Dear authors

Thank you for your submission. After careful reading of your revised manuscript, I feel that it needs further revision. You didn't address the concerns raised by reviewer 1. Hence I would like to see a revised draft with a detailed response to reviewer 1.

Reviewers' comments:

Reviewer's Responses to Questions

**Comments to the Author**

Reviewer #2: All comments have been addressed

2. Is the manuscript technically sound, and do the data support the conclusions?

Reviewer #2: Yes

3. Has the statistical analysis been performed appropriately and rigorously?

Reviewer #2: Yes

4. Have the authors made all data underlying the findings in their manuscript fully available?

Reviewer #2: Yes

5. Is the manuscript presented in an intelligible fashion and written in standard English?

Reviewer #2: Yes

Reviewer #2: (No Response)

**Do you want your identity to be public for this peer review?** For information about this choice, including consent withdrawal, please see our Privacy Policy

Reviewer #2: No

---

## [Author Response · Author response to Decision Letter 2]

1 Aug 2025

Response to Reviewer #1

1. Please cite a reference for the total viable method.

Thank you for the suggestion, we add the reference of the methodology of TVC on Line 87-88.

2. Line 81: Sampling procedure is not clear. Were five pieces (totally 25 g) cut and homogenized in 225 ml culture medium? Please add and explanation.

Thank you for your feedback, we have added more detailed information regarding sample preparation on Line 88-93.

3. Line 81-87: The composition of L-histidine broth is hardly understandable. Please clear. You may give the composition of artificial sea water in a separate sentence.

Thank you for the helpful suggestions, we rephrase the sentences of the L-histidine composition on the Line 91-93, and as for the Artificial seawater, the explanation added in a different sentence to be clearer on Line 93-97.

4. Lines 178-181: What are the means of Shannon and Simpson diversity indexes? Do temperature affect the diversity? Please discuss.

Thank you for the comments. As for the result of Shannon and Simpson diversity index, there were no significant difference which means the diversity and distribution of the samples were not different statistically (Line 218-225). This might indicates the variation in the diversity of bacteria between temperature treatments can be attributed to random chance rather than a true difference in the diversity of the samples.

5. Figure 2: There are significant differences between the replications in respect to microbial diversity. How could you explain this situation? Are these differences significant or expected? Please interpret these results in detail.

Thank you for the detailed comments. As shown in Figure 2, some variations were observed among the replicates. This is because different individual fish were used for each replicate. The fish were collected during different fishing seasons, and we anticipated potential changes in the bacterial community. However, this study focused on the dominant bacterial genera consistently present across all replicates. Therefore, we believe this does not pose a problem in interpreting the naturally occurring bacterial dominance, as there was no notable difference in dominant genera—Photobacterium was consistently observed in all replicates.

6. The conclusion parts needs to be improved. The last sentence (lines 250-251) has no contribution, since it has already been well known that the histamine production is related to microbial activity. Please mention that how these result can contribute to the safety of fish in respect to histamine.

Thank you for the constructive suggestion, which helped strengthen our conclusion. The final sentence was intended to emphasize the relationship between bacterial growth and histamine production, where higher bacterial growth observed at 10°C corresponded with greater histamine accumulation. We have revised this section (Lines 296–298) to improve clarity. Additionally, in response to your suggestion, we have expanded the concluding remarks (Lines 298–301) to highlight how this study focuses on naturally occurring bacteria in fish and demonstrates that histamine can still form despite minimal contamination. The conclusion now ends by underscoring the importance of temperature control as a critical measure for preventing histamine-related food poisoning in seafood.

7. The reference numbered 12 is too old. Please cite a recent reference.

Thank you for your comment. We have updated the references with more recent ones.

Response to Reviewer #2

General Comments

Line 70: The manuscript does not provide the number of mackerel samples used in the study and does not specify how they were distributed among storage temperature and time groups. Information on sample size and replication is critical for assessing experimental design, statistical validity, and reproducibility.

Thank you for your feedback, we have added more detailed information regarding sample preparation on Line 76-85.

Line 79: The authors state that total viable counts were performed with TSA agar. However, it is not clear from the text whether mesophilic, psychrophilic or histamine producing bacteria (HPB) were counted. It is stated that “HPB colonies were observed” but this is not supported by a specific identification method.

The incubation temperatures used are complicated and unclear: Incubation temperatures are given on TSA agar: 4°C, 20°C, 30°C. However, which group of bacteria do they target? For example: 4°C → psychrophilic 20–30°C → mesophilic

This relationship is not explained and it is not stated why these temperatures were chosen.

No selective media for histamine-producing bacteria (HPB): If HPB identification is to be made, selective media such as Niven’s agar, decarboxylase media would be required.

HPB cannot be identified with TSA alone, this is scientifically incorrect.

Thank you for the comments! This study employed four different temperature conditions: two for cold storage of the fish (4°C and 10°C) and two for incubation of bacterial cultures (15°C and 30°C). We add the cold treatments explanation in Line 82-85 and the bacteria incubation explanation in Line 101-103. In this study, cold storage temperatures were applied to evaluate the quality of mackerel during storage in terms of bacterial counts, bacterial diversity, and histamine accumulation. Meanwhile, the incubation temperatures were used to compare the survival and growth of psychrophilic and mesophilic bacteria that persisted throughout the storage period.

Selective media were not used for bacterial enumeration in this study, as the primary objective was not to specifically isolate histamine-producing bacteria, but rather to assess whether such bacteria are dominant among the natural bacterial population present on fresh mackerel. It was clearly established that the bacteria cultured on Tryptic Soy Agar (TSA) represent the naturally occurring microbiota of mackerel, not exclusively histamine-producing strains. In contrast, for the Most Probable Number (MPN) method, L-histidine was used as the primary substrate for histamine production. Bacterial isolates that successfully grew in this medium and subsequently tested positive for histamine production using paper chromatography were classified as suspected histamine-producing bacteria. This explanation added in Line 104-116.

Line 95-120, for all material and methods: PCR/amplicon sequencing part: "Amplicon sequencing" was performed, but which gene region was targeted (16S V3-V4 is understood from the table, but it is not clear in the text), which primer set, which control samples were used, where is the raw data?

Thank you for the detailed questions. In this study, the microbial community analyses using 16S rRNA sequencing with the targeted gene region V3–V4 region with binding sequence 410 bp and lead length of 280 bp. The primers for amplifying using a two-step tailed PCR are 341f and 805r with the detail in Table 1. As for the control samples, Negative control is prepared using nuclease-free water as a template. The revised method version was added in the Line 119-127. Raw data will be made available upon request.

The reader cannot clearly understand which bacteria were defined as HPB and on what basis. The methodology used for counting psychrophilic and mesophilic bacteria is not clear. TSA + ASW environment and temperatures are given, but which bacterial groups were classified as psychrophilic/mesophilic and on what basis is not explained.

Thank you for your concern regarding the classification basis of HPB! However, in this section, the specific identification of HPB was not conducted. Overall, the primary objective of this study was not to guide readers toward the precise identification of HPB, but rather to describe the naturally occurring bacterial community present in fresh mackerel during cold storage. Indeed, the results of this study showed that the dominant genus was Photobacterium, which has been previously identified in the literature as a psychrophilic bacterium responsible for histamine production. Therefore, based on both prior research and the findings of this study, Photobacterium is considered the primary suspected psychrophilic HPB.The methods for enumerating psychrophilic and mesophilic bacteria, as well as the use of TSA medium supplemented with ASW, have been explained and added in the methodological section above (Line 101-103; 114-116).

HPLC method: Reference is provided, but separation conditions, detector wavelength, calibration curves, sample preparation steps are quite superficial.

Thank you for your toughful comments! We have provided more technical information regarding the HPLC method on Line 137-152.

Aim: The novelty aspect is not emphasized enough: How is this study different from previous studies on histamine production? For example, was a new HPB strain identified at the metagenomic level?

Thank you for your valuable comment. We appreciate your interest in clarifying the novelty of our study. While the identification of a new HPB strain at the metagenomic level was not the primary focus of this work, our study offers several novel contributions:

• Minimization of external contamination

Unlike many previous studies that analyzed fish samples from retail or distribution points, our study used freshly caught mackerel, allowing us to investigate the naturally occurring bacterial community under controlled conditions with minimal external contamination. This approach provides clearer insights into the intrinsic bacterial dynamics of fresh fish during storage.

• Temperature-specific focus on psychrophilic HPB

We specifically explored bacterial behavior at 4°C and 10°C to reflect realistic cold-chain scenarios and to evaluate the role of psychrophilic bacteria in histamine accumulation. This temperature-focused analysis is less emphasized in prior literature, which typically assesses mesophilic HPB under broader or higher temperature ranges.

We have stated the novelty point in the cover letter and the manuscript to better emphasize these aspects of novelty and distinction from previous studies in Line 67-71.

Statistical analysis: The statistical analysis is limited to ANOVA. Correlation analyses between microbial diversity and histamine levels could have been performed.

Thank you for your valuable input. We have included a correlation analysis between bacteria number and histamine levels in Line 275-281 with a supplementary figure (S1_fig).

Moreover, the type of variance analysis applied is not specified in the manuscript.

We have now clarified in the revised manuscript that One-Way ANOVA (Single Factor) was used to analyze the differences in alpha diversity metrics (Shannon index, Simpson index, number of ASVs, and total read counts) between the 10°C and 4°C storage conditions in Line 156-158.

Figures and tables: The figures and tables are overly simplistic and not reader-friendly. For instance, Figure 2 presents only genus-level relative abundances; there is no specific analysis at the species level or of HPB carrying the hdc gene.

Thank you for your input, the reason we did not include species-level classification in this study is because amplicons sequencing targets fragments less than or equal to 400 bp of the 16S rRNA gene (V3-V4 region), which does not provide sufficient resolution for accurate and reliable species-level identification. For reference, previous studies have shown that 16S rRNA sequencing with short reads can lead to misclassification at the species level due to high sequence similarity among closely related taxa (Johnson et al., 2019; Janda & Abbott, 2007). Thus, we limited our analysis to the genus level to ensure greater confidence and accuracy in the interpretation of microbial community composition.

The manuscript should be thoroughly revised and rewritten by incorporating the above suggestions to present a clear, methodologically, and scientifically valuable study. Without these substantial corrections, the manuscript remains too ambiguous, inconsistent, and incomplete to be considered for publication in its current form.

Thank you once again for your time and professional review work on our manuscript. If there are any other modifications we could make, we would like very much to modify them and we really appreciate your help. We look forward to your feedback on the revised version.

References:

Janda, J. M., & Abbott, S. L. (2007). 16S rRNA gene sequencing for bacterial identification in the diagnostic laboratory: Pluses, perils, and pitfalls. Journal of Clinical Microbiology, 45(9), 2761–2764. https://doi.org/10.1128/JCM.01228-07

Johnson, J. S., et al. (2019). Evaluation of 16S rRNA gene sequencing for species and strain-level microbiome analysis. Nature Communications, 10, 5029. https://doi.org/10.1038/s41467-019-13036-1

---

## [Decision Letter · Decision Letter 2]

14 Aug 2025

Investigation of bacterial community and histamine production in fresh mackerel at low temperature storage

PONE-D-25-15404R2

Dear Dr. Takahashi,

We’re pleased to inform you that your manuscript has been judged scientifically suitable for publication and will be formally accepted for publication once it meets all outstanding technical requirements.

Kind regards,

Satheesh Sathianeson, Ph.D

Academic Editor

PLOS ONE

Additional Editor Comments (optional):

Reviewers' comments:

Reviewer's Responses to Questions

**Comments to the Author**

Reviewer #1: All comments have been addressed

2. Is the manuscript technically sound, and do the data support the conclusions?

Reviewer #1: Yes

3. Has the statistical analysis been performed appropriately and rigorously?

Reviewer #1: Yes

4. Have the authors made all data underlying the findings in their manuscript fully available?

Reviewer #1: Yes

5. Is the manuscript presented in an intelligible fashion and written in standard English?

Reviewer #1: Yes

Reviewer #1: The revised version is satisfactory and addresses the previous concerns; I have no further remarks or suggestions.

**Do you want your identity to be public for this peer review?** For information about this choice, including consent withdrawal, please see our Privacy Policy

Reviewer #1: No

---

## [Editor Report · Acceptance letter]

PONE-D-25-15404R2

PLOS ONE

Dear Dr. Takahashi,

I'm pleased to inform you that your manuscript has been deemed suitable for publication in PLOS ONE. Congratulations! Your manuscript is now being handed over to our production team.

Kind regards,

on behalf of

Dr. Satheesh Sathianeson

Academic Editor

PLOS ONE